# Physical Activity and Quality of Life among People with Intellectual Disabilities: The Role of Gender and the Practice Characteristics

**DOI:** 10.3390/bs13090773

**Published:** 2023-09-15

**Authors:** Evelia Franco, Carmen Ocete, Elena Pérez-Calzado, Ana Berástegui

**Affiliations:** 1Communication and Education Department, Loyola Andalucía University, Av. de las Universidades, 2, 41704 Dos Hermanas, Spain; efranco@uloyola.es; 2Education, Research Methods and Evaluation Department, Comillas Pontifical University, C/Universidad de Comillas 3, 28049 Madrid, Spain; epcalzado@comillas.edu; 3University Family Institute, Comillas Pontifical University, C/Universidad de Comillas 3, 28049 Madrid, Spain; a.berastegui@comillas.edu

**Keywords:** quality of life, physical activity, sports, intellectual disability

## Abstract

Staying physically active is synonymous with good health and well-being, and its benefits on the health of people with intellectual disabilities (PWIDs) have been studied. However, there is a lack of information on how it can influence their quality of life (QoL). This study aimed to analyze the relationship between QoL and physical activity in PWIDs according to gender and the characteristics of the practiced activity. A questionnaire was administered to 380 PWIDs (mean age of 28.23 ± 12.53), 54.21% of which were men. The QoL dimensions and second-order factors were studied in relation to practicing physical activity, the practice type, the context, and gender. The results indicate that people who practiced any activity showed better QoL values. Those who engaged in a nonregulated physical activity had better values in self-determination, emotional well-being, social inclusion, and personal development, while those who did sports presented higher scores in interpersonal relationships and physical well-being. In addition, it also appears that the association between physical activity and the QoL dimensions is distinct in inclusive and specific contexts. In conclusion, physical activity is related to a better QoL, although the impact of the practice type and context differs for each QoL dimension.

## 1. Introduction

Quality of life (QoL) is the state of personal well-being experienced when one’s basic needs have been met and when valuable aspects of life have been achieved and developed in different domains [1,2]. This variable encompasses multiple factors relevant for all people regardless of age or condition, although they manifest differently. Thus, it has etic—universal—and emic—cultural—properties. Furthermore, the construct comprises subjective and objective elements, and a person’s perception is essential for their QoL [3]. As a bridge between an individual’s personal life and the fulfillment of their social rights, QoL is a valuable tool to understand and enhance people with intellectual disabilities’ (PWIDs) equity, empowerment, and life satisfaction [1]. In Spain, according to the data collected by the national database of people with disabilities [4], there was a total of 274,833 people with a recognized intellectual disability (ID) in Spain at the end of 2018 (with a disability certificate with a degree equal to or greater than 33%). The improvement of QoL among this population closely aligns with the objectives of the CRPD—Convention on the Rights of Persons with Disabilities [5]—including the promotion, protection, and achievement of equality concerning all the human rights and fundamental liberties of disabled persons as well as promoting respect for their inherent dignity. This might explain why QoL has become a relevant concept and a methodology for public policy and why it supports development.

The multidimensional approach to quality of life proposed by Schalock and Verdugo [2,6] and Verdugo et al. [7] has been the most widely used approach in international research and consists of eight dimensions: personal development, which includes access to opportunities for learning and knowledge, training, and obtaining skills for self-realization; self-determination, referring to terms such as autonomy, privacy, and making decisions about objectives and goals that directly affect the individual; interpersonal relations, related to sharing emotions and having rewarding relationships with family, peers, friends, and romantic partners; social inclusion, referring to the person’s opportunity to receive social support as well as to participate and feel accepted and included in their social environment; rights, including the knowledge of their human rights and the promotion of their roles and responsibilities as citizens; emotional well-being, including feelings of happiness, security, and self-esteem as well as reduced daily stress; physical well-being, related to healthy habits, access to healthcare services, and a good physical shape; and material well-being, referring to economic resources, including housing, food, socioeconomic status, employment, and material goods. These eight dimensions can be grouped into three major factors named independence (personal development and self-determination), social (interpersonal relationships, participation, and rights), and well-being (emotional, physical, and material well-being) [7].

The QoL construct has recently been merged with the supports construct to create the quality of life supports model, highlighting the importance of providing people with disabilities with the necessary strategies and resources to enhance their QoL [8,9]. In this vein, the study of QoL has acquired an essential role in research since it helps to identify the needs of this population; the barriers to their well-being and inclusion; and the adequate policies, supports, and organizations needed to tackle them [9].

This explains the growing interest in understanding what factors and contexts may facilitate QoL in this population [10,11]. Many factors have been related to the QoL of PWIDs, including the personal characteristics of the nature and etiology of their disability [10], the level of their intellectual disability or support needs [11,12], and their personal goals and desires [10] as well as the impact of various support modalities, such as the type of schooling or the impact of work or independent living [12], on the size of the support organization, etc.

Recently, research has paid attention to the importance of physical activity in promoting the QoL of PWIDs, the benefits of physical activity on QoL being widely accepted [13]. Research indicates that people with disabilities have a 16–62% lower probability of adhering to the suggested physical activity recommendations, thereby increasing their vulnerability to the onset of severe health issues [14]. This situation is even more daunting among women. The current data indicate that only 3.488 out of the 13.051 people with disabilities with a federal license in Spain are women, which represents less than a quarter of all the sportspeople with disabilities in the country and 1% of all the women with disabilities in Spain aged between 15 and 49 [4]. These facts are especially worrying given that, as in the general population, inactivity and sedentarism are determining factors that contribute to adverse health in this population [15]. Over the last few decades, numerous studies have emphasized the positive effects of physical activity on the general health [16,17], cardiovascular fitness, musculoskeletal fitness, cardiometabolic risk factors, and brain and mental health outcomes of PWIDs [14]. However, fewer studies go beyond health-related QoL and focus on the overall impact and the benefits that physical activity can have on the life of PWIDs [18,19,20]. Despite the scarcity of studies in the literature, it has been suggested that physical activity might be related to the three major factors identified in QoL studies [2,6]: the independence, social, and well-being factors [21]. Regarding independence, it has been found that higher levels of physical activity are related to responsibility [22], independence, and self-determination [20]. Regarding the social factor, the literature points out the potential of physical activity to promote social inclusion [18]. Finally, the dimension that seems to receive the strongest impact from physical activity is well-being, including the physical, emotional, and material dimensions [21,23].

The impact of physical activity on the QoL of PWIDs may be affected by various factors, including the practice type and context. The practice type could be either a nonregulated physical activity or sports. On the one hand, according to Devís and Peiró [24], a nonregulated physical activity is understood as any conscious movement of the body executed with the skeletal muscles which involves energy consumption and facilitates interaction with the environment and the other living beings present in it. This physical activity includes the activities necessary for the maintenance of life, such as walking, and activities with recreational, health, or performance objectives, such as running or strength training in the gym [25]. On the other hand, a sport is defined as a physical activity with institutionalized rules [26]. In general, studies addressing physical activity and QoL in PWIDs have considered the different manifestations of this construct indistinctly and have thus ignored the potential differences in terms of the outcomes associated with both physical activity and sports [27].

The practice context is another factor that may condition the association between physical activity and QoL in this population. The existence of two different approaches for the participation of people with disabilities in physical activity contexts is widely accepted: the specific and inclusive approaches. The first one refers to a physical activity experience, usually a sport, adapted to the needs of people with disabilities or special health conditions [28]. In these contexts, people with disabilities exclusively interact with people with disabilities. In contrast, the inclusive approach seeks to enable people with disabilities to participate actively and effectively in any physical activity at the maximum level that their abilities allow [29]. This practice context involves people with and without disabilities. The concept of inclusion has received much attention in recent years, especially in educational contexts [30,31,32,33] and sports performance contexts [34]. There is currently a worldwide trend to create inclusive sports contexts, as this seems to favor the social inclusion of people with disabilities [35,36] and to contribute to their personal development [34].

### The Present Study

Considering the above information, this study aimed to advance the knowledge regarding physical activity and QoL among PWIDs.

Firstly, this study aimed to explore the potential differences in physical activity trends according to gender. According to the previous literature, it was hypothesized that men would be more likely to engage in physical activity than women. Also, men were expected to be more likely to engage in sports than women, who may advocate for other types of physical activity. No evidence was found regarding men and women’s preferences for practicing physical activity in specific or inclusive contexts, and no hypothesis could be formulated in this regard.

Secondly, this study sought to explore the association between physical activity and QoL. For this purpose, the differences in the QoL dimensions between those PWIDs who engaged in any physical activity and those who did not were studied. Likewise, the influence or absence of influence of the practice of physical activity on QoL was analyzed. According to previous works which have pointed out an association between the practice of physical activity and QoL, it was hypothesized that those engaging in some physical activity would report higher levels of QoL.

As a third objective, this study aimed to analyze how QoL among PWIDs who practiced physical activity differed between those engaged in a nonregulated physical activity and those who participated in sports. It is important to note that studies have yet to be carried out on PWIDs to better understand the association between the practice type (a nonregulated or sport-based practice type) and QoL among this population. The different nature of these two physical activity manifestations in terms of physical demands, interactions, and other aspects could lead participants to experience different levels of QoL. However, given the lack of evidence in this research line, no hypothesis could be formulated pointing to a specific direction.

Lastly, the present study sought to explore the potential differences in QoL between those PWIDs who practiced physical activity in specific contexts and those who did it in inclusive environments. Based on the existing scientific evidence, it was expected that PWIDs who participated in physical activity in inclusive settings might report higher levels of QoL than those who were engaged in specific physical activity contexts, especially in the social dimensions of QoL.

## 2. Materials and Methods

### 2.1. Design and Participants

A prospective cross-sectional design was used in this study. Nonprobabilistic convenience sampling was carried out following the procedure described in the Procedure section. To calculate sample size, the G*Power software [37] was used with the following parameters for mean difference analyses: two-tailed test, alpha error of 0.05, power of 0.95, effect size of 0.5, and allocation ratio in groups of 1. These parameters indicated a required sample size of 210 participants. The authors decided to collect at least 300 responses. The sample consisted of 380 participants, 54.21% of which were men (*n* = 206), 44.47% of which were women (*n* = 169), and 1.32% (*n* = 5) of which preferred not to state their gender, aged between 3 and 65 years (M = 28.23; SD = 12.53). In total, 48.4% (*n* = 184) of the participants self-completed the questionnaire, while proxies completed 51.5% (*n* = 196) of the questionnaires, which were parents in most cases.

All the participants had intellectual disabilities. Around 85% reported secondary diagnoses, Down Syndrome (*n* = 108; 28.4%), and borderline intellectual functioning (*n* = 47; 12.4%), which were the most frequent conditions. The remaining participants (44.2%) had other secondary diagnoses (e.g., 5.5% had autism spectrum disorder; 4.5% had epilepsy; and 3.2% had attention-deficit disorder/attention-deficit hyperactivity disorder). In terms of the percentage of disability issued by the Spanish Public Administration, 69 participants (18.2%) were given a disability degree between 33% and 64%, 164 participants (43.2%) presented a disability degree between 65% and 74%, and 71 participants (19.7%) were issued a disability degree greater than 75%. A total of 76 participants (20%) reported not knowing their disability degree.

### 2.2. Instruments

The questionnaire was designed with two versions: self-reported and reported-by-proxy versions. The self-reported version was checked for easy reading standards. Both versions included information on the following:

Sociodemographic variables. Participants were asked about their age, their gender, their percentage of disability, and diagnoses associated with their disability.

Physical activity practice. They were asked about their physical activity or sporting activity, the practice type (physical activity or sport), and the practice context (specific or inclusive).

Quality of life. The Comprehensive Assessment of the QoL of People with Intellectual or Developmental Disabilities scale (INICO-FEAPS) [3] was used. The tool consists of 72 items referring to eight dimensions: self-determination, rights, emotional well-being, social inclusion, personal development, interpersonal relationships, material well-being, and physical well-being. All dimensions comprise nine items answered on a Likert-type scale from 1 to 4 (1 corresponding to “never”, 2 corresponding to “sometimes”, 3 corresponding to “often”, and 4 corresponding to “usually”). In the study by Verdugo et al. [3], their Cronbach’s alpha values ranged between 0.53 and 0.80, being lower for the self-reported version. In the present study, Cronbach’s alpha ranged between 0.29 and 0.70. Given that internal consistency might have been compromised, it was decided to omit in the analyses the two dimensions whose values were below those found in the validation study of the tool (0.53). The two discarded dimensions were rights and material well-being. The factor structure showed adequate fit indices for a first-order structure of six dimensions (χ2 = 700.647, *p* < 0.05, CFI = 0.91, IFI = 0.91, and SRMR = 0.06).

### 2.3. Procedure

The questionnaire was distributed online among different organizations that work with PWIDs. Overall, more than 2000 potential participants and/or some proxies were contacted. Some of those PWIDs and proxies were listed in more than one database out of those accessed through collaborating institutions. Considering this and due to data protection issues, the exact number of people reached is unknown. A letter was sent to all entities explaining the aim of the study and asking them to disseminate the questionnaire online among their users. Likewise, collaborating with different community services in the Community of Madrid was requested. In this case, the questionnaires were administered by one research team member, who visited the occupational centers for PWIDs (settings were designed to enable occupational, personal, and social development and to thus favor their social and occupational integration) who agreed to participate in this study. The questionnaires were collected between November 2022 and May 2023, with an approximate application time of 30 min. Before completing the questionnaire, all participants were informed about the project and their rights as participants in the study (particularly, the right to review material and the right to withdraw from the process), and they signed an informed consent form. The work followed the guidelines of the ethical principles of the Declaration of Helsinki.

### 2.4. Data Analysis

Firstly, frequencies for physical activity participation were calculated. More precisely, the frequency of participants in any physical activity type, frequency of practice type, and frequency of practice context were determined. In this first step, a series of χ2 tests were also performed to test potential differences in frequency according to participants’ gender.

Descriptive statistics were calculated for the QoL dimensions according to the activity of the participants, practice type, and practice context.

The distribution of scores of the study variables was checked using the Kolmogorov–Smirnov test, and the need for nonparametric tests was found (*p* < 0.05). Nonparametric tests for independent samples (Mann–Whitney U test) were carried out to determine whether there were differences between groups, and the effect size of the differences was calculated using Cliff’s delta. According to Vargha et al. [38], values between 0.11 and 0.28 were considered small; values between 0.29 and 0.42 were interpreted as medium; and values higher than 0.42 were considered large. Given the scale used for the instrument measuring QoL (1–4), multivariate logistic regression analyses were carried out to determine the possible influence of the practice of physical–sport activity on the QoL of PWIDs. For this purpose, responses for QoL dimensions (dependent variables) were recorded in the database as 0 (scores below 2) and 1 (scores above 2). On the other hand, the practice of physical–sport activity (independent variable), considered to be a categorical variable, was created as a dummy variable. Sex was included as a confounding variable in the analysis. All analyses were conducted using SPSS 24.0 statistical package.

## 3. Results

### 3.1. Participation in Physical–Sport Activity according to Gender

In total, 75.3% (*n* = 286) of the participants practiced some physical–sport activity, while 24.7% (*n* = 94) reported not practicing any physical–sport activity. Of those who were engaged in physical–sport activity, 34.6% (*n* = 99) practiced individual sports; 35% (*n* = 100) were involved in team sports; and 30.1% (*n* = 86) practiced some other nonregulated physical activity. On the other hand, among the total number of practitioners, 35.3% (*n* = 134) practiced only with people with disabilities; 32.9% (*n* = 125) practiced inclusively with people with and/or without disabilities; and 5.3% (*n* = 20) practiced alone or with their family. Only 1.1% reported other forms of companionship during their physical–sport activity. Table 1 shows the distribution of the participants in terms of their physical–sport activity practice according to gender.

A Pearson’s χ2 test and an analysis of the adjusted residuals revealed that the ratio of participants who practiced any physical activity was not significantly different between men and women. As for the practice type, men were more likely to participate in sports, while women were more likely to participate in other nonregulated physical activity contexts. Lastly, no differences were found in the practice context between men and women.

### 3.2. QoL and Practice of Physical–Sport Activity

This section presents the data of the mean differences between the groups as well as those of the multivariate logistic regression analyses using the different dimensions of QoL as the dependent variable and physical activity as the independent variable.

Table 2 shows the descriptive statistics of the analyzed QoL variables and the Mann–Whitney U test results for the groups of those who practiced and those who did not practice physical–sport activity. The participants who practiced some physical–sport activity had higher QoL values in all the analyzed variables than those who did not. Statistically significant differences with high effect sizes were found for all the variables, except for the social factor, which had a medium effect size.

Table 3 displays the findings of the multivariate logistic regression analysis. It can be observed that the practice of physical–sport activity had a significant influence on the QoL variables, except for personal development, interpersonal relationships, physical well-being and general quality of life.

### 3.3. Differences in QoL According to the Practice Type

In this case, the differences in QoL were calculated according to the practice type, splitting the sample into those who practiced sports and those who were engaged in a nonregulated physical activity.

In general, the reported values were higher among those participants engaged in a nonregulated physical activity for all the QoL variables, except for interpersonal relationships and physical well-being in which the participants in sports displayed higher values (see Table 4). While no significant differences emerged in the general QoL variable or the three second-order factors (the independence, social, and well-being factors), significant differences in favor of the participants in nonregulated contexts of physical activity were found in emotional well-being, social inclusion, and personal development, these differences being medium.

### 3.4. Differences in QoL According to the Practice Context

Table 5 shows the descriptive data for the variables analyzed according to the practice context (specific or inclusive) of the physical–sport activity. People who practiced with a specific partnership presented significantly higher values in physical well-being, although these differences were small. Conversely, people who practiced inclusively presented significantly higher values in self-determination and emotional well-being, these differences being small to moderate. Furthermore, they showed higher scores in the second-order independence factor, this difference being small.

## 4. Discussion

The present study contributes to understanding QoL in PWIDs according to the physical–sport activity that they practice through four different and complementary aims.

### 4.1. Physical–Sport Activity and Gender in PWIDs

The first objective was to explore the potential differences in physical–sport activity trends according to gender. The hypotheses for this first aim were partially confirmed. Surprisingly, while expected, no association was found between gender and physical–sport activity practice. This is to say that there were no significant differences in the proportion of men and women engaged in physical–sport activity. As it had been hypothesized, there was an association between gender and the practice type (sports vs. nonregulated physical activities). Men were more likely to engage in sports, while women seemed to prefer other types of physical activity. This is aligned with the study by Anderson, Wozencroft, and Bedini [39], which suggests that the challenges in the participation of girls and women with disabilities likely include a lack of social support for girls with disabilities to participate in sports, and with Ascondo et al.’s study [40], which stated that women perceived more barriers than men in terms of practicing sports. A commonly used argument is that women and girls with disabilities are historically disenfranchised from physical–sport activity practice due to “double discrimination”: being female and having a disability. In turn, the absence of accurate data on the physical–sport habits of women with disabilities means that it is difficult to define the current state of the issue in question. An example of this is that, in Spain, the only figures related to the practice of sports by women with disabilities are the number of professional sports licenses and their participation in international competitions such as the Paralympics Games [41].

According to Rauner et al. [42], there seem to be gender differences in physical activity practice between men and women without a disability. This coincides with the study by Dėdelė, Chebotarova, and Miškinytė [43] on the difference in participation in physical activity between women and men; in this case, with disabilities, women showed lower levels of participation. Gender differences are found in the perception of barriers to physical activity, highlighting the importance of encouraging women with disabilities to participate in physical activity. In this line, the study by Gordon, Bloom, and Taylor [44] considers that women with disabilities are more susceptible to experiencing discrimination and exclusion compared to their male counterparts. The comparison of women with and without disabilities in the work of Sundahl et al. [45] showed that adolescent and young-adult women with intellectual disabilities had a significantly lower level of physical activity than women of the same age without an intellectual disability. This can be explained because women without intellectual disabilities more frequently had a leisure activity program with which they performed physical activity. This is in line with Quezada, Otaola, and Huete [46] in their work to find out the reality and needs of women with intellectual disabilities–Down Syndrome in Spain; they highlighted the need to promote the possibilities of practicing sports, especially in the stage of adolescence, as a measure to combat social isolation, highlighting the need to establish empowerment strategies for women with intellectual disabilities that generate sports social models. We agree with Olasagasti-Ibargoien et al. [47] that ways to overcome these barriers should be explored and that the most effective strategies to increase the participation of women with disabilities in physical–sport activity should be discussed.

There was no association between gender and the practice context (specific vs. inclusive). This finding suggests that gender might not be a determinative factor to engage in either the specific or inclusive practice contexts, and, therefore, other reasons better explaining why PWIDs opt to join a specific or inclusive group (such as their activity preference or the available opportunities) might exist. It is worth noting, though, that the number of male participants was slightly higher in specific contexts. This might be due to the fact that, as mentioned earlier, male participants are more likely to participate in sports, which, unlike other types of practice, are linked with competition. It could be that creating inclusive competitive contexts is a more challenging task than designing and offering other possibilities for physical activity practice. It would be interesting to activate mechanisms to increase the offer of inclusive competitive practices. This would be of great value not only for PWIDs but also for people without disabilities who could benefit from the experience of having contact with PWIDs. As suggested by previous works, being in contact with people with disabilities might positively impact the knowledge and awareness of people without disabilities towards this population [48,49].

### 4.2. Physical–Sport Activity and Quality of Life in PWIDs

The rest of the aims in this study focus on QoL and the physical–sport activity of PWIDs. Regarding the second objective, it was found that PWIDs who practiced any physical–sport activity displayed higher levels of QoL in all the analyzed dimensions, interpersonal relationships being the only dimension that did not show statistical significance. In our sample, the magnitude of the differences between those who were sedentary and those who were physically active were large. The largest differences were in the general well-being factor in accordance with the previous literature [21,27,50], the differences being larger in emotional well-being than in physical well-being, although these differences were still large. Likewise, the results show that there was an influence of the practice of physical–sport activity on all the dimensions of quality of life, except for the social dimension. This suggests that, although the importance of physical activity for physical well-being is crucial [14,17], its benefits for emotional well-being, including behavior and emotional problems, mental health, and psychosocial well-being, should not be underestimated and need further research attention [23,51].

Differences were also prominent in the general factor of independence and the subdimension of self-determination. This may contradict the idea that a person with a disability is not interested in being physically active [52], and it is aligned with previous results associating physical activity with greater responsibility [22], independence, and self-determination [21]. Physical activity can be a lever for self-determination for people with disabilities since it can promote making choices and can improve different skills and competencies [53].

Finally, the QoL factor that showed differences of a lesser magnitude was the social factor, again, as in the study by Carbó-Carreté et al. [21], partly because, although significant differences were found in social inclusion in agreement with the previous literature [18], no differences appeared in interpersonal relationships. The study by Blick et al. [18] suggested that PWIDs who practiced some physical activity reported having more frequent outings into the community and, therefore, having better social inclusion or engagement than their inactive peers. On the contrary, they did not find differences in family satisfaction, a key component of interpersonal relationships.

As for the third objective, differences in the QoL dimensions were found between the practitioners engaged in sport activity and those who practiced another type of physical activity. More specifically, it was found that PWIDs who practiced a nonregulated physical activity presented higher values in self-determination, emotional well-being, social inclusion, and personal development. In contrast, those who practiced sport activity had higher scores in the interpersonal relationships and physical well-being dimensions. These findings suggest that either nonregulated physical activities or sports can improve QoL among PWIDs. Interestingly, each of them enhances different QoL dimensions. On the one hand, participating in a nonregulated physical activity might foster the personal and social dimensions (such as self-determination and personal development) better than a sport itself. The reasons why people engaged in a nonregulated physical activity scored higher than those participating in sports need to be clarified, and no studies have been found addressing this research question. It could be that, when participating in certain nonregulated physical activities, such as walking or cycling, participants might feel they make more decisions and are more autonomous (e.g., to choose their routes, company, pace, etc.) than when practicing a sport with a coach instructing them on how to perform. According to the self-determination theory [54], it is possible that, within this context, participants are more likely to feel that their needs for autonomy and competence are fulfilled, which could result in higher self-determination and more positive development [55,56]. To better understand the differences in emotional well-being, future studies must deepen the analysis by considering how different physical activity manifestations affect emotional states. On the other hand and in line with the present study’s findings, previous works have suggested that sports can be a valid tool to improve interpersonal relationships [57]. Due to their nature, people who practice sports usually join a sports club, which might foster positive social experiences [58,59]. The present findings also align with the existing evidence of an association between sports and physical well-being [16,54]. The fact that sports are unavoidably linked to competition might lead to more demanding experiences in terms of physical requirements, which could explain why those engaged in sports reported higher levels of physical well-being. It would be interesting for future studies to gain a better understanding of the role that competition might play in QoL among PWIDs. Furthermore, it is worth pointing out that the proxies’ perception could differ from the participants’ views when it comes to analyzing the potential of these two different contexts. Future studies could further explore this research line.

Lastly, this study explored whether QoL perceptions differed between PWIDs who practiced physical activity in a specific context and those who did it in an inclusive context. The findings in the present study revealed that people who practiced with a specific partnership had higher values in physical well-being. In contrast, people who practiced inclusively had higher self-determination and emotional well-being scores. Also, this group had better values in independence. We agree with what has been stated by Hansen et al. [60] in their study to identify factors that improve the odds of success for longitudinal participation in physical activity for people with intellectual disabilities. In this study, the social contextual factors became determinant and even more critical when obtaining information about these people’s perceptions, motivations, and belief systems. Therefore, it is essential to establish a dialogue and to include PWIDs in the process to understand their behaviors, described as needs, preferences, and perceived barriers.

In the last two decades, an incipient interest has emerged regarding the inclusion of people with disabilities in general sports contexts. The fostering of inclusive physical activity, recreation, and sports that entail the participation of people with and without disabilities is a common global trend [61]. The complexity of this multifactorial process has yet to yield any evidence on the effect on PWIDs when comparing their participation in specific and inclusive sports contexts. The participation of people with disabilities in general sports contexts is understood as an expression of inclusion. This implies that attending to the context and, therefore, to the impact on the person with disabilities is a crucial aspect of participation. In that sense, the practice context of a physical–sport activity has a differential effect on QoL. While inclusive contexts are related to greater self-determination and emotional well-being, specific physical–sports contexts are related to greater physical well-being. The specialization of sports technicians and adapting the practice to the unique needs of PWIDs are likely associated with this practice’s more significant physical benefit. In contrast, emotional and personal benefits are more related to the inclusive approach.

### 4.3. Limitations and Future Lines

Despite making a relevant contribution to understanding QoL and physical–sport activity among PWIDs, this study had limitations. Firstly, it must be pointed out that the results cannot be generalized, given the convenience sampling method that was used. Also, the heterogeneity of the sample must be noted since the participants were significantly distinct in terms of their age and support needs. There were 10 participants who were under 10 years old, and, in all of these cases, the responses were provided by their parents. It would be advisable for future studies to explore whether the role that physical–sport activity plays on the QoL of PWIDs varies according to these (age and support needs) and other sample characteristics. In this line, multiple informants during data collection (PWIDs and proxies) might have impacted the QoL scores as previously suggested [62]. A second limitation was the cross-sectional design, which prevented us from testing causal hypotheses about the impact of physical–sport activity on QoL. Quasi-experimental studies are needed to better understand the causality of this relationship among PWIDs. Lastly, although some physical–sport activity characteristics were considered in the analyses (the practice type and practice context), many others should receive some attention, such as the quantity/frequency of the physical–sport activity. Future research should incorporate these variables through objective measures (e.g., accelerometers or step counters).

### 4.4. Practical Implications and Conclusions

This study is an interesting contribution to the understanding of QoL among PWIDs. More specifically, it adds to the existing literature by approaching physical activity and sports as potential contexts for fostering QoL in this population.

The results obtained in this study suggest that PWID centers and services devoted to enhancing their QoL should include any manifestation of physical–sport activity in their plannings and programs regardless of the gender of the practitioners using both their organizations and the resources available in the community. However, it is possible to consider the difference found between genders in terms of their preferred practice type, knowing that, generally, women are more likely to practice and generate more adherence to nonregulated physical activities. At the same time, men do so towards regulated sports. In addition, to promote the physical–sport activity practice among PWIDs of the female gender, another measure that the institutions involved could implement would be establishing institutional networks that generate solid structures that are stable over time and that are aimed at developing opportunities for inclusive practice.

Likewise, it would be advisable to know the interests and needs of PWIDs before prescribing the type of physical activity (a nonregulated physical activity or sport). Although both manifestations favor dimensions of QoL, a nonregulated physical activity could favor personal dimensions, such as self-determination or personal development. At the same time, sports seem to be more beneficial for improving physical well-being.

Finally, inclusive contexts of practice should be generated to contribute to this population’s holistic development, promoting different aspects that affect their QoL, such as self-determination, emotional well-being, and independence. These applications can provide information on which to base the generation of national public policies adapted to the needs and current situation of PWIDs in Spain.

Overall, these findings highlight the potential of physical activity and sports to promote PWIDs’ integration and social inclusion.

## Figures and Tables

**Table 1 behavsci-13-00773-t001:** Distribution of participants among the different physical–sport-activity-related study variables.

	Men	Women	(d.f) Pearson’s χ2
Physical–sport activity practice			
Yes	153 (74.3%)	129 (76.3%)	(2) 0.273
No	53 (25.7%)	40 (23.7%)	
Practice type			
Nonregulated physical activity	33 (21.6%)	52 (40.3%)	(2) 11.74 **
Sport	120 (78.4%)	77 (59.7%)	
Practice context *			
Specific	78 (51.3%)	54 (42.5%)	(6) 4.78
Inclusive	65 (42.8%)	58 (45.7%)	

* Participants who responded “on my own/with my family” or “others” were included in this table. ** *p* < 0.005.

**Table 2 behavsci-13-00773-t002:** Descriptive statistics and Mann–Whitney U test results for the QoL variables as a function of sport–physical activity practice.

Quality of Life Variables	Practices PSA (*n* = 286) M (SD)	Does Not practice PSA (*n* = 94) M (SD)	Z	*p*	ES
Self-determination	2.89 (0.56)	2.49 (0.53)	−6.27	<0.001	0.73
Emotional well-being	2.38 (0.44)	2.07 (0.44)	−5.7	<0.001	0.70
Social inclusion	2.49 (0.44)	2.26 (0.38)	−4.47	<0.001	0.54
Personal development	2.87 (0.55)	2.60 (0.48)	−4.21	<0.001	0.50
Interpersonal relationships	2.79 (0.49)	2.76 (0.50)	−0.70	0.482	0.05
Physical well-being	2.89 (0.39)	2.68 (0.37)	−4.74	<0.001	0.55
Independence factor	2.88 (0.50)	2.54 (0.45)	−5.92	<0.001	0.69
Social factor	2.64 (0.38)	2.51 (0.35)	−2.87	0.004	0.34
Well-being factor	2.64 (0.33)	2.38 (0.30)	−6.90	<0.001	0.81
General quality of life	2.81 (0.34)	2.56 (0.31)	−6.82	<0.001	0.77

*p* < 0.05; ES = effect size.

**Table 3 behavsci-13-00773-t003:** Multivariate logistic regression analysis for the QoL variables as a function of sport–physical activity practice.

Quality of Life Variables	β	SE	Wald Statistics	*p*	AOR	Nagelkerke Pseudo R^2^
Self-determination	1.18	0.49	5.87	0.02	1.25–8.51	0.5
sex	0.32	0.48	0.43	0.51	0.53–3.54	
Emotional well-being	1.51	0.28	28.74	<0.001	2.60–7.81	0.12
sex	0.002	0.26	0.00	0.99	0.61–1.65	
Social inclusion	1.27	0.36	11.39	<0.001	1.7–7.40	0.07
sex	−0.08	0.35	0.05	0.82	0.46–1.84	
Personal development	0.71	0.58	1.49	0.22	0.65–6.41	0.02
sex	0.32	0.56	0.32	0.57	0.46–4.12	
Interpersonal relationships	−0.27	0.66	0.17	0.68	0.21–2.77	0.01
sex	0.58	0.54	1.12	0.29	0.61–5.17	
Physical well-being	1.14	0.83	1.90	0.17	0.62–15.75	0.03
sex	−0.13	0.78	0.03	0.87	0.19–4.04	
Independence factor	1.57	0.66	5.68	0.02	1.32–17.46	0.08
sex	0.69	0.69	0.99	0.32	0.51–7.68	
Social factor	1.15	0.59	3.76	0.05	0.99–10.02	0.05
sex	0.51	0.61	0.69	0.41	0.50–5.48	
Well-being factor	1.74	0.64	7.38	0.007	1.62–19.85	0.09
sex	0.02	0.60	0.001	0.97	0.32–3.27	
General quality of life	1.13	1.01	1.25	0.26	0.43–22.19	0.03
sex	−0.14	0.95	0.02	0.89	0.14–5.64	

Note: In the line where the name of the QoL dimension appears, the regression values for sport–physical activity are presented. *p* < 0.05.

**Table 4 behavsci-13-00773-t004:** Descriptive statistics and Mann–Whitney U test for QoL variables as a function of the type of physical–sport activity.

Quality of Life Variables	Nonregulated PA (*n* = 86) M (SD)	Sport (*n* = 200) M (SD)	Z	*p*	ES
Self-determination	2.94 (0.55)	2.86 (0.56)	−0.98	0.325	0.10
Emotional well-being	2.48 (0.44)	2.34 (0.44)	−2.69	0.007	0.32
Social inclusion	2.59 (0.49)	2.45 (0.41)	−2.37	0.018	0.31
Personal development	2.98 (0.54)	2.82 (0.55)	−2.31	0.021	0.29
Interpersonal relationships	2.76 (0.50)	2.79 (0.49)	−0.37	0.711	0.06
Physical well-being	2.89 (0.40)	2.90 (0.38)	−0.39	0.699	0.02
Independence factor	2.96 (0.48)	2.85 (0.51)	−1.91	0.056	0.22
Social factor	2.68 (0.40)	2.62 (0.37)	−1.18	0.239	0.14
Well-being factor	2.68 (0.34)	2.62 (0.32)	−1.59	0.111	0.20
General quality of life	2.87 (0.35)	2.79 (0.34)	−1.84	0.066	0.21

*p* < 0.05; ES = effect size.

**Table 5 behavsci-13-00773-t005:** Descriptive statistics and Mann–Whitney U test for QoL variables as a function of the partnership.

Quality of Life Variables	Specific (*n* = 134) M (SD)	Inclusive (*n* = 125) M (SD)	Z	*p*	ES
Self-determination	2.79 (0.54)	2.97 (0.58)	−2.580	0.010	0.32
Emotional well-being	2.32 (0.44)	2.42 (0.43)	−2.044	0.041	0.25
Social inclusion	2.43 (0.43)	2.52 (0.42)	−1.670	0.095	0.20
Personal development	2.80 (0.57)	2.90 (0.52)	−1.431	0.152	0.18
Interpersonal relationships	2.79 (0.45)	2.78 (0.51)	−0.463	0.643	0.02
Physical well-being	2.94 (0.38)	2.85 (0.40)	−2.193	0.028	0.23
Independence factor	2.80 (0.50)	2.93 (0.49)	−2.26	0.024	0.28
Social factor	2.61 (0.36)	2.65 (0.40)	−0.63	0.529	0.10
Well-being factor	2.63 (0.33)	2.64 (0.32)	−0.31	0.759	0.02
General quality of life	2.78 (0.35)	2.84 (0.34)	−1.282	0.200	0.15

*p* < 0.05; ES = effect size.

## Data Availability

The data presented in this study are available on request from the corresponding author. The data are not publicly available due to data protection restrictions.

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
