# Peer review of "Physical Activity and Quality of Life among People with Intellectual Disabilities: The Role of Gender and the Practice Characteristics"

_behavsci, 2023, doi:10.3390/bs13090773_

Round 1
Reviewer 1 Report
The following items are suggested for further revision.
1. Line 152, the number of sample recruited and the number of returned questionnaires didn’t report.
2. Line 155 and 444, over 50% proxies to complete the questionnaire. Without indication and reference from literature about the validity and reliability of this arrangement, the results of the study becomes non-trustable. This is a key consideration in conducting questionnaire study for people with mental impairment. Author may need to clarify if the strange results (Line 281, Table 3) in this study is not affected by the proxy arrangement.
3. Line 285, typo error on table number.
4. Line 309 and 351, Results indicated that no association between gender and practice context but the discussion on this issue was limited.
5. Line 385, 386 – didn’t discuss or address why non-regulated physical activity favor for self-determination, emotional wellbeing, social inclusion and personal development.
Reviewer 2 Report
The article can be published, the topic is of scientific and current interest and is close to the special issue of Behavioral Sciences Journal and I congratulate you for this. Regarding the elaboration of the manuscript, in my opinion, there are several proposals for improvement:
The objective of the study should be more clearly defined in the first part of the paper, bringing arguments regarding the necessity of these researches, with reference to improving the quality of life (QoL) of people with disabilities. I would recommend arguing the importance of the study by defining the concept of disability in national legislation, presenting some statistical data (children, young people, adults, elderly with disabilities in Spain), because it would help to understand the specificity and the national context in Spain, because there are different procedures and problems in the intercultural measurement of the QoL. The need for these studies would also be useful from the perspective of integration and educational and social inclusion, especially for children and young people; I recommend a longer bibliography, the analysis of other articles, such as:
https://doi.org/10.3390/su13137056
https://doi.org/10.3390/su13084545
https://doi.org/10.3390/ijerph17030931
Research methodology: the research methodology should bring additional information regarding the population that participated in the study, the method of selection of participants; it is not clear how the research sample was selected. However, the following sections, where the study is described and conclusions are drawn, have a fair design, well structured and cohesive. Perhaps a more detailed presentation of the multivariate research designs used to assess the ways in which personal characteristics and environmental variables relate to person-rated QoL would be useful. Presentation of a section referring to the limitations of the study, demonstrates honesty and frankness in the investigation, but also implications for future research. In my opinion, this section is very convenient and timely.
The conclusions of the study should present the importance and necessity of these researches for the QoL of people with disabilities, their integration and social inclusion; also, the conclusions of the study should be analyzed from the perspective of the results of other studies; references to more studies.
Reviewer 3 Report
The Article Title: Physical Activity and Quality of Life among People with intellectual disabilities: the role of Gender and practice characteristicsby Evelia Franco , Carmen Ocete , Elena Pérez-Calzado , Ana Berástegu The study aim is to analyze the relationship between QoL and physical activity in PWIDs according to gender and the characteristics of the practiced activity. The results indicate that people who practiced any activity show better QoL values.
Reviewer 4 Report
Dear Authors,
Thank you for the opportunity to review the manuscript „Physical Activity and Quality of Life among People with intellectual disabilities: the role of Gender and practice characteristics“.
Personally, I was very interested in reading this manuscript.
Although this study is relevant, I have major concerns:
1. What was the design of this study?
2. Which analytical variables were dependent and independent?
3. How was the representative sample size calculated for this study?
4. Lines 120-121: the authors wrote: “Considering the above, this study aims to advance knowledge regarding physical 1activity among PWIDs and the association between such activity and QoL“.
However, the analysis of empirical data was performed only to reveal differences between the investigated groups. I would suggest that additional statistical tests must be applied (e.g. using regression analysis or structural equation modeling (SEM)) in respect to clarify the association between the physical activity and the QoL.
5. The minimum age of subjects according to the information on line 154 was 3 years. If parents completed questionnaires for the subjects, did this not distort the accuracy of the data in this study?
6. Why has the data in this study been analysed by gender (Table 1)?
7. The conclusions of this study do not fulfil the specific aims of the study. Undoubtedly, the design of this observational research is a cross sectional study. I recommend that the authors set out dependent and independent variables very clearly; clearly formulate the aims and/or research questions (RQ) of the present study; adapt the required statistical analysis methods (and re-analyze data) in order to test deductive hypotheses to a sufficient level.
8. In addition, other limitations of this study are related to the lack of availability to generalize the results of the study, as the convenience sampling method was used.
Kind Regards
Round 2
Reviewer 1 Report
The concerns about proxy arrangement was addressed. Reader may aware this issue accordingly.
Author Response
Thank you for your comment.
Reviewer 4 Report
Dear Authors,
Thank you for the manuscript changes.
I have just one concern and suggestion:
As you are not able to generalize data, thus those do not make any sense. Both internal validity and external validity of this study were not maintained.
Furthermore, you did not perform logistic regression analyses along with predictive model (with adjustments for confounders). Finally, you can not make any recommendations or implications. This is a big mistake. I can not help with this question.
However,
I wish the authors great success.
Author Response
Thank you for your comments, we hope we have responded to them with the changes we have made.
Please see the attachment.

Round 3
Reviewer 4 Report
Dear Authors,
Thank you for the opportunity to re-review the topic “Physical Activity and Quality of Life among People with intellectual disabilities: the role of Gender and practice characteristics “.
I have next observations as follows:
First, the single cross-sectional study in design was carried out.
The present study explored potential differences in QoL between those PWIDs who practice physical activity in specific contexts and those who do it in inclusive environments.
Therefore, dependent variables were related to QqL.
All dimensions of QqL comprised nine items with the possible answers on Likert-type scale from 1 to 4 (1 corresponding to “never,” 2 to “sometimes,” 3 to “often,” and 4 to “usually”). Hence, you can not perform linear regression analyses (in terms of four type of answers; a scale must be equal to five of those ones). Therefore, according to cut-off points the authors may classify and recode the scores (2 > median ≥ 2) derived at the answers for each dimension.
Independent variables were related to the levels of physical activity.
Authors wrote “Physical activity practice. They were asked about their physical activity or sporting activity, the practice type (physical activity or sport), and the practice context (specific or inclusive) (lines 195-196)”.
The results section contains analysis by this independent variable (practice physical activity vs. not practice physical activity as reference category), right.
Secondly, you revealed the differences in the groups of males and females. This is good finding, which could serve as a confounder in your regression models.
Finally, according to dependent and independent variables, the authors may perform the prognostic multivariate logistic regression models, which must be adjusted for sex (confounder) of study participants. Furthermore, the authors must show beta values ± SE (standard errors), Wald statistics, p-value, and adjusted odds ratios (AOR) as well as Pseudo-R-squared values (in terms of Nagelkerke R2 ≥ 0.2) in all descriptions of each logistic regression model.
Also, the authors can use other statistical analysis methods such as Factor analysis or Structural equation modelling (SEM). It must be highlighted that the authors performed bivariate analysis, however it does not make sense if it comes together with single cross-sectional study in design.
After all calculations and multivariate analysis, you will disclose the results more accurately. This type of analysis will be more friendly to the deductive hypothesis have been tested. Given that the results will change, my suggestion is associated with following adjustments of discussion section, too.
Kind Regards
